# Multiracial Race Self-Labeling Decisions: The Influence of Gender, Social Class, and Political Party Affiliation

Sarah Elizabeth Castillo 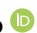

Sociology, University of Tennessee Knoxville, Knoxville, TN 37803, USA; scastil1@vols.utk.edu

**Abstract:** The growing prominence of the multiracial population in the United States is prompting new questions about the importance of social identities on race self-labeling decisions. I review and expand on a growing body of research on this population that focuses on identifying and describing nonracial categories important to shaping racial identities. Specifically, I utilized a national survey of U.S. adults administered by the Pew Research Center to investigate how social identities defined by nonracial categories such as gender, social class, and political party affiliation impact the race self-labels of multiracial people. In addition, I consider factors of racial identity, discrimination, and social pressure and their potential influence on race self-labeling decisions. The findings indicate that gender, social class, and political party affiliation are potential predictors of race self-labeling decisions of multiracial people. After adding the factors of racial identity, discrimination, and social pressure, the results remain significant. In addition, the results for social class and political party affiliation reinforce the actuality that a pervasive racial hierarchy and social stratification system, situated in the context of White supremacy, is embedded within U.S. society.

**Keywords:** multiracial; race; gender; social class; political party affiliation

## 1. Introduction

In 2000, the federal government allowed people to select more than one race category on their US Census forms (Jones and Smith 2001). That year, approximately 2.4 percent of the US population self-identified with two or more races, and a decade later that number grew to 2.9 percent (Jones and Smith 2001; Jones et al. 2021). Fast forward another decade to the 2020 census, and the multiracial population had grown to an astounding 10.2 percent (33.8 million people) of the total population. The most significant change was seen for the White and other race racial combination, adding 17.6 million people to the total multiracial population, signifying a 1000 percent change. People reporting more than one race on the census changed more than all single-race groups (Jones et al. 2021). Race reporting on the US Census has significant implications for government funding and social support systems. The distribution of USD 675 billion in federal funding to local, state, and tribal governments is informed by US Census data. The US Census determines the distribution of congressional seats to states and how states and communities allocate funding for neighborhood improvement, education, and public health (United States Census Bureau 2018). In addition, the US Census is used as a political device to assess the social and cultural discourse regarding race (Bratter 2018; Davenport 2016b; Rockquemore et al. 2009).

The confidentiality of US Census forms provides an opportunity for agency in race-labeling decisions that multiracial people lack in person-to-person interactions (Albuja et al. 2017; Bratter 2018; Saperstein and Penner 2012; Shih and Sanchez 2009). Multiracial people's cultural understanding of race, values, and beliefs shape their race category decisions (Albuja et al. 2017; Bratter 2018; Saperstein and Penner 2012; Shih and Sanchez 2009). For example, claiming a racial group identity could be indicative of multiracial people's political affiliation or vice versa (Conover 1984). Political party affiliation is often tied to a sense of culture, values, and moral convictions (Davenport 2016a; Hochschild and

Weaver 2007; Weaver 2012). How a person identifies politically depends on their shared group meanings and access to the resources that result from group membership (Davenport 2016a; Hochschild and Weaver 2007; Rockquemore et al. 2009; Weaver 2012). Multiracial political identities have the potential to reinforce or challenge the existing racialized social structures (Rockquemore et al. 2009). Coupled with the limitations of accurate self-race reporting on the US Census and the potential impact of the multiracial population on the concept of race itself, it is becoming important to understand the factors, such as political party affiliation, that influence race self-labeling decisions of multiracial people.

I investigated the influence of gender, social class, and political party affiliation on nonpublic race self-labeling decisions of multiracial Americans. I also investigated the influence of additional factors, such as racial identity, discrimination, and social pressure. Assessing the private labeling decisions of multiracial people, with and without White racial backgrounds, provides insight on how nonracial categories such as gender, social class, and political party affiliation inform personal understandings of race, situated within the context of White supremacy that is embedded within U.S. society.

### 1.1. Theoretical Framework

I approached this project through the lens of critical race theory, intersectionality theory, and social identity theory. Utilizing these theories provided a framework in which to understand and/or identify the impact of multiple social forces on multiracial race self-labeling decisions.

### 1.1.1. Critical Race Theory

Critical race theory consists of a collection of activists and scholars that are interested in the relationship among race, racism, and power (Delgado and Stefancic 2012). It operates on the premise that race is not a natural or biologically grounded way in which to distinguish between groups of human beings. Instead, race is a socially constructed classification that is used to maintain a system of racial stratification that oppresses and exploits people of color. This is accomplished through the normalization of racism within legal, social, economic, and political institutions (Bonilla-Silva 1997, 2018; Delgado and Stefancic 2012; Feagin 2006; Feagin and Ducey 2018).

Critical race theory illustrates how society operates as a system of white-over-color ascendancy that serves both a psychic and material purpose, specifically for the dominant group and the perpetuation of colorblind ideology. White supremacy rests at the center of this relationship and is a legal, cultural, and political condition that is reproduced and maintained. Primarily, the system of race can be divided into two codependent parts: one of outright racism and the other White privilege. Through these avenues Whites help and promote their interests and places of power (Bonilla-Silva 1997, 2018; Delgado and Stefancic 2012; Feagin 2006; Feagin and Ducey 2018).

According to critical race theory, racism is a pervasive, systemic, and deeply ingrained mechanism of society (Delgado and Stefancic 2012). There are four common forms of prevalent racism that impact the social construction and understanding of race. Internalized racism lies with the individual and their ideology. Interpersonal racism occurs between individuals and is what often comes to mind when we first think about racism. Institutional racism occurs within institutions and systems of power such as the legal system or the education system. Finally, there is structural racism, in which racial bias is perpetuated across society through social agents and systems. Racism advances the interests of both White elites and the White working class through the concept of interest convergence, and large segments of society, often with considerable power, have little incentive to change (Bonilla-Silva 1997, 2018; Delgado and Stefancic 2012; Feagin 2006; Feagin and Ducey 2018).

Ultimately, critical race theory emphasizes how race, racism, and the associated power socially construct White privilege. In this context, it would be expected for multiracial people to choose a White race self-label given the choice to do so. A desire to meet the standards of society, to gain power and privilege, and avoid negative interactions are

fundamental desires. Living in a White supremacist society and facing several forms of racism on a regular basis must impact race self-labeling decisions for multiracial people.

### 1.1.2. Social Identity Theory

According to social identity theory, a person's sense of who they are is based on their group membership (Tajfel 1982; Tajfel and Turner 1979). People have multiple identities that are activated by different social contexts and emphasized by category membership, which provide a sense of pride and self-esteem (Burke and Stets 2009; Stets and Burke 2000; Tajfel 1982; Tajfel and Turner 1979). The self is seen as a reflexive object that is categorized, classified, and self-named in relation to group membership that in turn provides a sense of common identification with a collective social category (Stets and Burke 2000; Tajfel 1982; Tajfel and Turner 1979; Turner and Reynolds 2011).

In addition, collective identity provides access to desired group resources (Stets and Cast 2007). Social identity theory conceptualizes a resource as both entities and social processes that sustain a system of social interaction associated with group membership (Stets and Cast 2007; Tajfel 1982; Tajfel and Turner 1979). Resources assist people in accomplishing goals and obtaining desired effects in social interaction (Stets and Cast 2007; Tajfel 1982; Tajfel and Turner 1979). Access to resources can influence the salience of a social identity in a given social setting through the motivation of self-enhancement and uncertainty reduction (Stets and Cast 2007).

Social identity theory facilitates the examination of the implications and motivations of social identities for personal decisions. The action and agency behind a collective identity provides a mechanism to portray and sustain a shared system of values and beliefs (Hogg 2018; Tajfel 1982; Tajfel and Turner 1979). Access to resources associated with social identities both supports and recruits group membership (Stets and Cast 2007; Tajfel 1982; Tajfel and Turner 1979). Insights into the motivations and benefits of social identities help us to understand how these identities influence personal decisions such as race self-labeling decisions. People have multiple social identities of varying salience that can be activated in any given social situation (Stryker 1968). It is crucial to further examine how these identities influence racial self-labeling, and social identity theory serves as an important guide in that process.

### 1.1.3. Intersectionality Theory

Intersectionality theory provides additional justification for including such factors as gender, social class, and political party affiliation in the examination of racial self-labeling. Specifically, this theoretical perspective argues that multiple social positions influence identity choices (Hill Collins and Bilge 2016). According to Patricia Hill Collins and Bilge (2016, p. 115), intersectionality theory is defined as a "multifaceted perspective acknowledging the richness of multiple socially constructed identities that combine to create each of us as unique individuals". Different combinations of social categories create different experiences with structure, culture, and personal interactions in society (Hill Collins and Bilge 2016). Mutually constructed identity categories resulting from shared intersections enable the development of a collective identity because social categories related to group memberships are not mutually exclusive but instead build on each other and work together (Hill Collins and Bilge 2016).

Through the lens of intersectionality, it is possible to see how self-race labels of multiracial people do not merely describe a person's racial identity; instead, they represent a specific social stance that is used to justify the values and beliefs associated with a collective identity (Hill Collins and Bilge 2016). Black men experience the social world differently than White women. Black women experience the social world differently than Asian women. Group membership identities are not mutually exclusive; instead, they build upon each other (Hill Collins and Bilge 2016). Navigating the structural and cultural complexities of intersecting identities is difficult in and of itself when we look at binary relationships between distinguished social categories such as race and gender. Multiracial people have

the added complexity of racial identity intersections that create unique experiences based on the social context and interaction with the social structure. The added layer of a racial identity intersection provides an avenue for agency in favoring one or another racial identity to suit a situation. Through the intersections of social identities, multiracial people have the ability to sustain or challenge the social organization of power in relation to race (Hill Collins and Bilge 2016).

### 1.2. Gender

Multiracial women and men are likely to make different race-labeling decisions based on their gender experiences. Research links women's appearance to their sense of group acceptance and rejection more so than for men (Hunter 2005; Rockquemore and Brunsma 2008). For instance, Kerry Ann Rockquemore (2002) found that multiracial women experienced negative interactions with Black women because of their appearance. The negative interactions centered on the physical appearance of the multiracial women and how they did not exhibit characteristics of what it meant to be Black, even partially Black. The respondents' Black portion of their identity was invalidated by the Black women and validated by other non-Black people, creating issues with identity for the multiracial respondents (Rockquemore 2002). Beauty standards act as a social resource and status marker for multiracial women, making them less likely to be perceived as racial minorities (Hunter 2007). Women experience pressure to perform gender in a specific manner that emulates racist standards of beauty (Hunter 2005; Rockquemore and Brunsma 2008). This gives rise to colorism, which is a form of discrimination centered on Western hegemonic beauty standards. Colorism is a type of skin tone stratification that maintains preferential treatment for people with lighter skin over people with darker skin and affects all genders (Hunter 2005, 2007; Strmic-Pawl 2014).

Research has also found that biracial men are more easily accepted by minority peers than their female counterparts (Hunter 2005, 2007). Because men are accepted and validated by their minority peers, they are more likely to associate with that racial group (Butler-Sweet 2017; Hunter 2005, 2007; Rockquemore 2002). However, biracial women often experience hostility from their minority peers and are perceived as a social threat, specifically in what these studies refer to as the mating market (Butler-Sweet 2017; Hunter 2005; Rockquemore 2002). Research suggests that there is a perceived shortage of eligible Black males for Black females, and they in turn consider biracial women as competition that personifies both minority status and White beauty standards associated with colorism (Butler-Sweet 2017; Hunter 2005; Rockquemore 2002). Colorism is a stratification system that works to divide people in order to maintain a system of White dominance (Hamilton et al. 2009; Hersch 2011; Hunter 2002, 2005; Thompson and Keith 2001).

Overall, research reports that women experience higher levels of behavior and phenotype invalidation in social interactions than men do (Franco and O'Brien 2018; Hunter 2005, 2007). A potential explanation for this trend is that women are perceived as less threatening than men and as easier targets for discussion and/or actions (Franco and O'Brien 2018; Hunter 2005, 2007). Although research suggests that biracial identity has emerged from a failure of acceptance from other race groups, gender might not have significant implications for racial identity choice (Rockquemore and Brunsma 2008). At the same time, gender is believed to structure the identity process through the different experiences of men and women (Hunter 2005, 2007; Rockquemore and Brunsma 2008).

Group membership with gender is much more difficult to change than other group memberships. The effects of racial and gender biases are more difficult to avoid, as people must navigate the racial hierarchy and the gender hierarchy at the same time (Penner and Saperstein 2013). The contextual nature of race identification is magnified in combination with gender and highlights the intersectionality of other social categories (Albuja et al. 2017; Hill Collins and Bilge 2016; Hunter 2005; Rockquemore and Brunsma 2008).

*1.3. Social Class*

Not unlike the impact of gender, multiracial adults are likely to make race self-labeling decisions based on their social class. A pervasive racial hierarchy and social stratification system is embedded within the social class system (Lei and Bodenhausen 2017; Strmic-Pawl 2014). Research shows that social class might have implications for racial identification (Lei and Bodenhausen 2017). Specifically, the category of "poor people" is mentally represented as relatively Black or minority, and the category "rich people" is mentally represented as White (Lei and Bodenhausen 2017; Penner and Saperstein 2013). Relatedly, prior research has suggested that people experiencing economic hardships become more likely to connect with marginalized ethnic or racial groups and those that escape economic hardships do not (Penner and Saperstein 2013; Simonovits and Kezdi 2016). The interplay between race and poverty influences the likelihood or unlikelihood of a multiracial or minority race identification (Bratter 2018).

Colorism, as mentioned before, plays a significant role in the influence of social class on race-labeling decisions as well, through the concept of dominant group ethnicity (Doan 1997; Hunter 2005, 2007). In the United States the dominant racial group consists of White people, typically of Western European ancestry (Doan 1997). There is a hidden ethnicity present within this dominant group that creates a "whiteness standard" that affects social class (Doan 1997). For instance, economic and residential affluence associated with White ethnicity "whitens" racial identification (Davenport 2016b). Multiracial people that have the ability to "pass" as a White person may be more inclined to self-identify as White to obtain access to the privileges and resources of the dominant group (Bratter 2018; Davenport 2016b; Hunter 2005; Rockquemore and Brunsma 2008). The status associated with different levels of social class that is in turn reinforced by interactions with the dominant group ethnicity impacts race-labeling decisions (Bratter 2018; Davenport 2016b; Hunter 2005; Penner and Saperstein 2013; Thornhill 2015).

The enactment of race labels is contextual and can change over time as people navigate or possibly change their social status positions (Bratter 2018). Prior research suggests that across racial and social class backgrounds, people from higher-status groups are more likely to claim a biracial identity or a White identity than their counterparts from lower-status groups (Thornhill 2015; Townsend et al. 2012). There is an incentive for multiracial people to conform to a set standard of Whiteness that eases the concerns of the previously mentioned dominant ethnic group (Doan 1997; Thornhill 2015).

*1.4. Political Party Affiliation*

Similar to gender and social class, race self-labeling decisions of multiracial people can be impacted by their political party affiliations. Political party affiliation is often tied to a sense of culture, values, and moral convictions (Davenport 2016a; Hochschild and Weaver 2007; Weaver 2012). Prior research on multiracial political participation has focused on voting behavior trends and policy support (Khanna 2012). Political party preference research is growing but has attributed political affiliation choices to parental influence and shared group attitudes (Davenport 2016a; Hochschild and Weaver 2007; Weaver 2012). Multiracial political identities have the potential to reinforce or challenge the existing racialized social structures (Rockquemore et al. 2009). How a person identifies politically depends on their shared group meanings and access to the resources that result from group membership (Davenport 2016a; Hochschild and Weaver 2007; Rockquemore et al. 2009; Weaver 2012).

Research has found that the brain treats race and politics as coalitional alliances (Pietraszewski et al. 2015). In 2014, 49 percent of White people were Republican-Party-leaning and 40 percent Democratic-Party-leaning (Pew Research Center 2015). In contrast, 80 percent of Black people were Democratic-Party-leaning, and 11 percent were Republican-Party-leaning (Pew Research Center 2015). This leads to an observation that politics is significantly racially divided (Davenport 2016a; Pietraszewski et al. 2015).

The political party preference aspect of race self-labeling decisions is slightly broader-reaching than the other categories previously mentioned (Conover 1984; Davenport 2016a). Political leanings often support or oppose the current racial stratification system through the political environment. Specifically, Conover (1984) argues that people respond to the political world in terms of what is deemed important by politicians, parties, and media coverage. In addition, politics often mirror group interests and indirectly self-interests. Moreover, group identifications, such as race, represent a critical factor in determining how people perceive the political world. In other words, claiming a racial group identity could be indicative of a multiracial person's political affiliation or vice versa (Conover 1984).

*1.5. Racial Identity*

Race is typically conceptualized as a social construct that creates, maintains, and perpetuates a stratified racial social system (Albuja et al. 2017; Saperstein and Penner 2012; Shih and Sanchez 2009). Race labels have real, social, political, and economic consequences. Multiracial people can select a single race that resonates with them, or they can choose a multiracial identity. These choices are contextual and often change according to the expectations of social roles and group membership (Albuja et al. 2017; Harris and Sim 2002; Rockquemore and Brunsma 2008; Saperstein and Penner 2012). A person's self-understanding of what racial category they belong to is their racial identity (Rockquemore et al. 2009). This identity is influenced both by how people regard their racial composition or identity privately and how others regard their race publicly (Sellers et al. 1998). Opportunities such as surveys, job applications, census forms, and other private interactions with racial category selection have the potential to be good indicators of a multiracial person's racial identity (Rockquemore et al. 2009; Sellers et al. 1998).

Research suggests that biracial people may face harsh social evaluations due to the fluidity of their identities (Albuja et al. 2017; Harris and Sim 2002; Rockquemore and Brunsma 2008). To avoid social penalties, multiracial people may act in a manner associated with a specific race in a specific interaction. For instance, if a half-Black, half-White person is interacting with a group of friends that are primarily Black in a social setting, this person may behave in a manner that highlights his/her Black background. This can be as simple as preferred attire or the use of slang or language that is common within that group. In a different setting with primarily White people the same person may change the way they speak or the way they dress to garner acceptance. In essence, they act "White" when they are around White people, and they act "Black" when they are around Black people.

Self-race labeling outside of person-to-person interactions is similar to the social situations listed above (Albuja et al. 2017; Harris and Sim 2002; Rockquemore and Brunsma 2008). For instance, a half-Black and half-White person may select Black as a race option on a college scholarship application. The same person may select White on an application to join an affluent country club. The person decides what race they wish to be perceived as in each situation instead of what race is typically assigned to them by others in interpersonal relations.

Multiracial people select identities from a variety of racial choices that are influenced by the validation or lack of validation from other people in social interactions. Some scholars argue that there are four common racial identity options for multiracial people that include: singular, border (exclusively biracial), protean (sometimes Black, sometimes White, and sometimes biracial), and transcendent (no racial identity) (Harris and Sim 2002; Rockquemore and Brunsma 2008; Rockquemore et al. 2009). Findings indicate that the majority of respondents preferred the border identity. This was found to be the result of a constant process of validation that operates from a push-and-pull perspective (Harris and Sim 2002; Rockquemore and Brunsma 2008; Thornhill 2015). Negative responses or treatment from single-race peers push multiracial people away from identifying with that particular race and pulls them towards a biracial identity.

Research suggests that multiracial people may experience negative interactions with single-race people due to the fluid experience of their multiracial identities (Albuja et al. 2017; Harris and Sim 2002). This fluidity violates the established social norms of a stable race identification and can lead to negative social interactions. Multiracial people in turn learn or choose to enact a contextual racial presentation. They may privately identify in a certain way but regulate their public racial presentation according to the social situation (Albuja et al. 2017; Harris and Sim 2002). Findings suggest that even though the race presentation is regulated, it is still penalized by monoracial perceivers, especially White perceivers (Albuja et al. 2017; Harris and Sim 2002). However, other research found that awareness of being able to activate specific racial identities in certain social contexts provides social benefit for multiracial people (Gaither et al. 2013; Thornhill 2015).

Ownership, Privilege, and the Negotiation of Racial Identity

As mentioned above, multiracial people have a sense of fluidity for race self-label choices, and this is especially true if White is a part of their racial background. Whiteness in this sense can be thought of as property. A claim to Whiteness is connected to rights typically assigned to physical property, such as legal rights, rights of disposition, and rights of use and enjoyment. Ultimately, Whiteness is a valued social and national identity that is linked to power and privilege (Crenshaw et al. 1995; Delgado and Stefancic 2012).

Treating Whiteness as property sets the standard in many social situations and reinforces and perpetuates the power of White supremacy. Whiteness as being "normal" can be seen in the literature, movies, tv shows, and other forms of pop culture. Whiteness has a legal definition, primarily in the context of immigration law, but still legally creates a boundary between privilege and a lack of privilege (Crenshaw et al. 1995; Delgado and Stefancic 2012). At the end of the day, White privilege provides a myriad of social advantages that are associated with being a member of the dominant race.

It is not difficult to consider that multiracial people would desire to negotiate their race self-labeling choices in order to gain privilege or power. They may choose to identify or racially present themselves through what scholars refer to as abstract liberalism. This is the performance, typically among Whites, of appearing reasonable and racially moral while simultaneously opposing practical approaches to change in the racial system, often ignoring the reality of institutional and structural systems of racism (Bonilla-Silva 2018; Crenshaw et al. 1995; Delgado and Stefancic 2012). Taking on this stance is a form of racial negotiation. A multiracial person can appear to oppose the systems that perpetuate our racist society while at the same time reaping the benefits of the system by selecting a White identity.

*1.6. Discrimination*

Previous studies considered whether and how experiences with racism and discriminations might have implications for racial identity formation for multiracial people. For example, a study by Miville and colleagues (2005) found that encounters with racism and discrimination raised people' awareness of group membership with one race or another. In addition, these encounters helped people realize their uniqueness as a multiracial person and therefore created two types of identity experiences, one being a person of color and the other being a multiracial person. This study also demonstrated that multiracial people faced acts of racism in relation to both types of identity experiences (i.e., monoracial and multiracial). These experiences resulted in what Miville and colleagues (2005) refer to as the "chameleon effect", in which multiracial people select which identity to activate to reduce the likelihood of distress.

Colorism is also a prominent and powerful form of racial discrimination towards multiracial people (Hunter 2005, 2007; Rockquemore 2002; Rockquemore and Brunsma 2008). One notable result of colorism is exclusion from racial groups from both sides of the color spectrum (Butler-Sweet 2017; Hunter 2005). Light-skin multiracial people reported negative interactions with dark single-minority people, most often women, that led them to develop strong anti-Black and/or antiminority sentiments (Butler-Sweet 2017; Hunter 2005; Rockquemore 2002). Some of the interactions centered on racial identity with the lighter person being told that they were not racially authentic (Hunter 2005). Because the lighter person automatically received the privileges of being on the lighter side of the color spectrum, it was assumed that they embraced this view and could not claim an authentic minority identity (Butler-Sweet 2017; Hunter 2005; Rockquemore 2002). For multiracial respondents, it was assumed that since they were not completely "one" race, they could not possibly understand what it means to be a minority or dark-skin person. Respondents emphasized that they had to consciously work to legitimize their minority racial background (Butler-Sweet 2017; Hunter 2005; Rockquemore 2002).

### 1.7. Social Pressure and Location

It is also important to take into consideration the implication of social pressure from family and friends for the development of racial identity. Kerry Ann Rockquemore (2002) examined the family dynamics of multiracial people. Her results indicate that multiracial respondents that were raised by their White mothers frequently reported difficulty dealing with their White parent's explicit racism and racialized negativity towards their Black father. The same respondents experienced negative treatment from Black women in their community due to their multiracial status. Rockquemore (2002) indicates that identity formation is a process of external categorization, constraint, individual agency, and the negotiation of group interactions. Overall, these findings suggest that pressure from family and the contested relations between Black and multiracial people might make a difference in the development of racial identity and illustrate the importance and power of skin color (Rockquemore 2002).

At the same time, Davenport's (2016b) study indicates that family structure (i.e., single parent vs. married parents) might have little impact on racial identity and suggests that respondents preferred incorporating the race of both parents into self-identification. However, Davenport (2016b) found that neighborhood composition and region might be more predictive of racial identity. Namely, respondents were more likely to select a biracial identity as contact with their corresponding minority race increased in their neighborhood. Furthermore, research by Davenport (2016b) demonstrated some regional differences. Specifically, respondents living in the South were more likely to select a non-White racial identity and respondents in the Midwest tended to select a non-White or biracial identity, whereas respondents living in the Pacific West or Northeast were more likely to identify themselves as biracial. These findings suggest that neighborhood composition might be potentially related to social pressure and hence has an impact on race self-labeling decisions.

### 1.8. Nonracial Categories and Race Label Choices Current Study

On the basis of critical race theory, social identity theory, and prior research, I examined whether nonracial social categories such as gender, social class, and political party affiliation influence race self-labeling decisions of multiracial people. In addition, I investigated whether the importance of racial identity, the existence and experiences with discrimination, and social pressure have implications for race self-labeling and potentially impact the associations between nonracial social categories (i.e., gender, social class, and political party affiliation) and race self-labeling. I present the following three research questions

1.  Do differences in race self-labeling of multiracial people vary by the nonracial social categories of gender, social class, and political party affiliation?
2.  Can additional factors, such as racial identity, discrimination, and social pressure, be predictive of multiracial race self-labeling?
3.  Can additional factors (i.e., racial identity, discrimination, and social pressure) intercede the influence of nonracial social categories, (i.e., gender, social class, and political party affiliation) on multiracial race self-labeling?

## 2. Methods

To assess the effects of nonracial categories on race self-labeling decisions and identification of multiracial people, I used data from the Pew Research Center's Survey of Multiracial Americans conducted by the GfK Group using KnowledgePanel. KnowledgePanel members are a nationwide panel of participants recruited through RDD (Random Digital Dialing) and ABS (Address-Based Sampling) probability sampling methods. Panel members are recruited annually to account for panel attrition. The Survey of Multiracial Americans was conducted from 6 February 2015 to 6 April 2015 and was administered in both English and Spanish. The survey focused on identity, personal experience, and the social views of multiracial people in the United States.

Demographic information was collected from 21,224 adults nationwide in two stages. Stage one was a sample of general population adults as well as oversamples of non-Hispanic single-race Blacks and Asians that were identified using GfK's panelist profile data. The second stage consisted of a general population sample split randomly into four panel member replicates. A series of five screening questions were asked to all panelists to determine their racial background. To qualify as multiracial, a panelist must have selected two or more races for themselves, identified biological parents with different races, identified grandparents with different races, and/or identified great-grandparents with different races. In addition, Hispanic panelists were included if they met the same criteria previously listed and indicated that they consider Hispanic as a race (Pew Research Center 2018).

The dataset provides a qualifying filter, QFLAG (1 = respondents qualified and 2 = respondents did not qualify) that indicates whether a respondent qualified for the mixed-race portion of the survey. In addition, a mixed-race filter, XMIXED, was used in the following combinations: 1–7 when analyzing results from the general population survey and 8–11 when analyzing results based on the 21,224 adults that were screened for the multiracial survey and when analyzing results based on panelists that qualified for the survey. Last, another filter variable was provided that indicates the different ways in which a respondent could be considered mixed race based upon the family criteria listed above, FILTER (1 = respondents who selected more than two races for themselves, 2 = respondents considered multiracial based on their parents' race(s), 3 = respondents considered multiracial based upon grandparents' race(s), 4 = respondents considered multiracial based upon great-grandparents' race(s); 5–8 correspond to the first four options with the added consideration of Hispanic as a race.

Pew Research Center recommended the weighting of the data from the Survey of Multiracial Americans to adjust the results of studies that use this dataset. Doing so matches the data to the March 2014 Current Population Survey (CPS) in terms of the estimates on characteristics such as age, race, education, and language proficiency. In addition, individual racial and ethnic groups were weighted to be internally representative of age, gender, census region, metropolitan status, education, and household income (Pew Research Center 2018). For this study I used the provided WEIGHT2 option and the corresponding filtering combination, QFLAG 1, XMIXED 8–11, and FILTER 1, 2, 3, 5, 6, and 7, to obtain a sample of the population. I chose to not use respondents that qualified for the survey based on their great-grandparents' race(s) in order to simplify the inclusion criteria of multiracial status. In addition, I excluded missing value cases for the three main independent variables (gender, social class, and political party). This resulted in a final

sample size of *N* = 1564. All variables were based on the self-reports of adults aged 18 and older.

## 2.1. Dependent Variable

In this study, I assessed a single categorical indicator of race self-label. Respondents were asked to mark all categories that applied to them from a list of possible races, including White only, Black/African American only, Asian/Asian American only, American Indian only, Native Hawaiian or other Pacific Islander, Hispanic only—no races, 2 or more races, and some other race/refused. The Asian/Asian American only, American Indian only, Native Hawaiian or other Pacific Islander, and some other/refused response categories were omitted from the analysis due to their small sample size. This resulted in four race-self labeling options (*N* = 1430): White only (reference category), Black only, Hispanic only—no races, and multiracial.

## 2.2. Independent Variables

I examined three main independent variables in this study that captured responses for gender, social class, and political party affiliation. I examined three additional areas of independent variables that captured responses for racial identity, discrimination, and social pressure.

The first independent variable, gender (1 = male (reference category), 2 = female), captured the respondents' gender. The dataset did not provide for other gender options.

The second main independent variable was social class. Respondents were asked "If you were asked to use one of these commonly used names for the social classes, which would you say you belong in?" with the following response options: lower class, lower-middle class, middle class, upper-middle class, and upper class. To adjust for small response size, I collapsed the Lower and Lower-middle class categories into one category of lower-middle/lower class. I repeated this process for the upper-middle and upper-class categories. This resulted in a social class variable with three response options (1 = lower-middle/lower class, 2 = middle class, and 3 = upper-middle/upper class (reference category)).

The third main independent variable was political party affiliation. Respondents were asked "In politics today, do you consider yourself a..." with the following response options: Republican, Democrat, Independent, and something else. The something else category was recoded as missing and not used in the study. This resulted in a political party variable with three response options (1 = Republican (reference category), 2 = Democrat, and 3 = Independent).

### 2.2.1. Racial Identity Variables

To capture the concept of racial identity importance, I used the question, "Now we want you to think about your own personal identity, that is, the various ways that you define yourself as a person. How important are each of these characteristics to your own personal identity?". The characteristic that I chose to examine was race. The response options were: essential to your identity; important but not essential to your identity; and not too important. I reverse-coded the categories, resulting in a racial identity importance variable with three response options (1 = not important (reference category), 2 = important, and 3 = essential).

To capture respondents' racialized actions, I used the following question and response categories to create a composite measure: "Have you ever done any of the following things to try to influence how others see your race?" The questions provided the following "things" options: dressed in a certain way; talked in a certain way; worn your hair a certain way; and associated with certain people. The response options for each category were: yes, have done this and no, have not done this. The response options were reverse-coded. Based on these questions, I created a composite measure of racialized action (Cronbach's alpha = 0.819) variable with two response options (1 = no (reference category and 2 = yes).

2.2.2. Discrimination Variables

To take into account the influence of discrimination, I used two measures: respondents' opinions on the existence of discrimination and respondents' experiences with discrimination. To capture respondents' opinions on the existence of discrimination, I used the following question from the survey "How much discrimination do you think there is today against people in the United States who are of each of the following races or origins?" and a list of relevant subquestions on racial minority groups examined in the present study (i.e., Hispanic or Latino and Black or African American). The original response categories for these questions were: a lot of discrimination, some discrimination, only a little discrimination, and no discrimination. For the purposes of the present analysis, I created a composite measure using the responses for Hispanic or Latino and Black of African American categories. I created a continuous variable using the mean scores of the two questions. I then created a discrimination opinion (Cronbach's alpha = 0.760) variable with response options (1 = none to some discrimination and 2 = a lot of discrimination (reference category) to consider whether respondents perceived discrimination as existing against relative minority groups in the U.S.

To measure whether respondents had ever experienced discrimination, I created a composite measure of experienced discrimination (Cronbach's alpha = 0.752). I used the survey question "For each of the following, please indicate whether or not it has happened to you because of your racial background" and a list of related subquestions on potential types of discrimination, including: has been threatened; had been subject to slurs and jokes; has been treated unfairly by an employer in hiring, pay, or promotion; has been unfairly stopped by the police; and has received poor service in restaurants, hotels, or other places of business. The original response categories for these questions were: yes, has happened in the last 12 months; yes, has happened but not in the past 12 months; and no, has never happened. I collapsed the first two response categories, which resulted in an experienced discrimination variable with responses options (1 = no (reference category) and 2 = yes).

2.2.3. Social Pressure Variables

Based on the question, "Have you ever felt pressure to choose one of the races in your background over another from the following groups?" and a list of these social groups. I created three dichotomous measures of pressure from family members, friends, and society in general (1 = no (reference category) and 2 = yes). The original response categories were: yes, have felt pressure; no, have not felt pressure, sometimes, rarely, and never. For the purposes of the present analysis, I reverse-coded the original response categories. This resulted in three pressure variables of family pressure, friend pressure, and societal pressure.

To further consider respondents' experiences with social pressure in relation to the community, I used the following two questions. First, for friend networks, I used, "How many of your close friends are White?", and for neighborhood networks, I used, "How many of the people in your neighborhood are White?". Response options were: all of them, most of them, some of them, and none of them. For the purpose of this study and small response sizes, I collapsed the response options into a dichotomous variable (1 = none to some and 2 = most to all (reference category). This resulted in the White friends and White neighbors' variables.

*2.3. Control Variable*

I controlled for age as a continuous variable. Respondents were asked to enter their age in years.

### 2.4. Analytical Approach

I present descriptive statistics for all of the study variables in Table 1. I also conducted zero-order correlations to test for multicollinearity. The zero-order correlations did not indicate issues with multicollinearity because none of the correlations among the study variables exceeded 0.54. The Pearson's correlation for societal pressure and friend pressure 0.54 ($p < 0.001$) was the highest score although within accepted limits; but these two variables are measuring a similar concept (pressure to conform).

I used multinomial logistic regression analyses because my dependent variable, race self-labeling, was a nominal variable with more than two response categories. I started by examining the implications of my main independent variables (i.e., gender, social class, and political party affiliation) for race self-labeling (Table 2).

**Table 1.** Weighted Descriptive Statistics for Study Variables ($N = 1430$).

| Variables | Count | Weighted Frequency | Weighted % |
|---|---|---|---|
| **Race Self-Label** | | | |
| White Only (reference category) | 592 | 414.97 | 45.57 |
| Black Only | 204 | 242.26 | 26.60 |
| Hispanic Only—No Races | 108 | 87.25 | 9.58 |
| Multiracial | 526 | 166.14 | 18.24 |
| Female | 820 | 537.96 | 59.08 |
| **Social Class** | | | |
| Lower-Middle/Lower Class | 511 | 316.49 | 34.76 |
| Middle Class | 725 | 471.42 | 51.77 |
| Upper-Middle/Upper Class (reference category) | 194 | 122.70 | 13.47 |
| **Political Party Affiliation** | | | |
| Republican Party (reference category) | 339 | 201.06 | 22.08 |
| Democratic Party | 575 | 420.50 | 46.18 |
| Independent Party | 516 | 289.05 | 31.74 |
| **Racial Identity Importance** | | | |
| Essential to Identity | 327 | 225.71 | 32.12 |
| Important to Identity | 633 | 389.25 | 42.97 |
| Not Important to Identity (reference category) | 463 | 290.99 | 32.12 |
| Yes, Racialized Actions | 296 | 203.59 | 22.76 |
| **Discrimination Existence** | | | |
| A Lot of Discrimination Exists | 587 | 403.43 | 44.30 |
| Discrimination Experience | | | |
| Yes, Experienced Discrimination | 952 | 618.51 | 67.93 |
| **Social Pressure** | | | |
| Yes, Pressure from Family | 123 | 91.35 | 10.11 |
| Yes, Pressure from Friends | 122 | 86.94 | 9.60 |
| Yes, Pressure from Society | 239 | 151.37 | 16.74 |
| None to Some White Friends | 590 | 448.89 | 49.74 |
| None to Some White Neighbors | 569 | 424.32 | 47.19 |
| **Control Variable** | | | |
| Age [a] | | 54.91 | 16.22 |

[a] Reported mean and standard deviation. Note: Values were weighted using poststratification weights.

**Table 2.** Multinomial Logistic Regression Results: Gender, Social Class, and Political Party Affiliation (*N* = 1430).

| Variables | Black Only | Hispanic Only—No Races | Multiracial |
|:---:|:---:|:---:|:---:|
| Female | 1.25 | 0.88 | 0.64 * |
| Social Class [a] | | | |
| Lower-Middle/Lower Class | 1.32 | 0.75 | 0.72 |
| Middle Class | 0.91 | 0.52 | 0.98 |
| Political Party Affiliation [a] | | | |
| Democratic Party | 14.31 *** | 6.13 *** | 1.48 |
| Independent Party | 2.93 ** | 1.29 | 1.02 |
| Age | 1.00 | 1.01 | 0.98 *** |
| Constant | 0.07 | 0.08 | 1.22 |
| Wald chi-square (df) | 166.53 (18) | | |
| Pseudo $R^2$ | 0.13 | | |

[a] Reference category: White Only, Upper-Middle/Upper Class, Republican Party. * $p < 0.05$. ** $p < 0.01$. *** $p < 0.001$.

To investigate whether additional factors such as racial identity importance, discrimination, and social pressure are predictive of race self-labeling and can intercede the impact of nonracial social categories (i.e., gender, social class, and political affiliation) on race self-labeling, I included separate blocks of measures of these additional factors separately in Models 1–3 in Table 3. That is, Model 1 examined the implication of the importance of racial identity. Model 2 considered opinions on the existence of racial discrimination and experiences with racial discrimination. Model 3 considered the covariates of family pressure, friend pressure, societal pressure, friend networks, and neighborhood networks. Finally, I present the full model that considers all the covariates in Table 4. All the models controlled for respondents' age.

**Table 3.** Multinomial Logistic Regression Results: Racial Identity, Discrimination, and Social Pressure (*N* = 1430).

| | Model 1 | | | Model 2 | | | Model 3 | | |
|---|---|---|---|---|---|---|---|---|---|
| | Racial Identity Importance | | | Discrimination | | | Social Pressure | | |
| | Black Only | Hispanic Only No Races | Multiracial | Black Only | Hispanic Only No Races | Multiracial | Black Only | Hispanic Only No Races | Multiracial |
| Female | 1.48 * | 0.93 | 0.66 * | 1.47 * | 0.96 | 0.66 * | 1.14 | 0.88 | 0.64 * |
| Social Class [a] | | | | | | | | | |
| Lower-Middle/Lower Class | 1.22 | 0.84 | 0.75 | 1.40 | 0.74 | 0.71 | 0.87 | 0.52 | 0.69 |
| Middle Class | 0.95 | 0.59 | 1.03 | 0.93 | 0.53 | 0.98 | 0.47 * | 0.35 ** | 0.84 |
| Political Party Affiliation [a] | | | | | | | | | |
| Democratic Party | 12.21 *** | 5.82 *** | 1.38 | 8.65 *** | 5.29 *** | 1.39 | 10.57 *** | 4.98 *** | 1.44 |
| Independent Party | 2.81 ** | 1.40 | 1.03 | 2.11 * | 1.13 | 0.97 | 2.40 * | 1.17 | 1.08 |
| Racial Identity Importance [a] | | | | | | | | | |
| Essential to Identity | 3.59 *** | 2.40 * | 1.23 | | | | | | |
| Important to Identity | 1.91 ** | 2.79 ** | 1.18 | | | | | | |
| Yes, Racialized Actions | 2.42 *** | 1.25 | 1.66 * | | | | | | |
| A lot of Discrimination Exists | | | | 3.90 *** | 1.40 | 1.18 | | | |
| Yes, Experienced Discrimination | | | | 2.89 *** | 2.02 * | 1.21 | | | |
| Social Pressure | | | | | | | | | |
| Yes, Pressure from Family | | | | | | | 0.80 | 0.67 | 0.64 |
| Yes, Pressure from Friends | | | | | | | 0.77 | 0.342 | 0.42 |
| Yes, Pressure from Society | | | | | | | 3.37 ** | 7.09 *** | 3.00 ** |
| None to Some White Friends | | | | | | | 20.88 *** | 4.12 *** | 1.96 ** |
| None to Some White Neighbors | | | | | | | 1.56 * | 1.97 * | 0.97 |
| Age | 1.01 | 1.01 | 0.98 ** | 1.01 | 1.02 | 0.98 *** | 1.02 * | 1.02 ** | 0.98 ** |
| Constant | 0.03 | 0.04 | 0.91 | | | | | | |
| Wald chi-square (df) | 199.98 (27) | | | 214.06 (24) | | | 269.02 (33) | | |
| Pseudo $R^2$ | 0.17 | | | 0.18 | | | 0.28 | | |

[a] Reference category: White Only, Upper-Middle/Upper Class, Republican Party, and Not Important to Identity. * $p < 0.05$. ** $p < 0.01$. *** $p < 0.001$.

**Table 4.** Multinomial Logistic Regression Results: All Study Variables (*N* = 1430).

|  | **Black Only** | **Hispanic Only—No Races** | **Multiracial** |
|---|---|---|---|
| Female | 1.43 | 0.98 | 0.64 * |
| Social Class [a] |  |  |  |
| Lower-Middle/Lower Class | 0.85 | 0.59 | 0.72 |
| Middle Class | 0.48 * | 0.37 ** | 0.88 |
| Political Party Affiliation [a] |  |  |  |
| Democratic Party | 6.19 *** | 4.45 *** | 1.29 |
| Independent Party | 1.96 | 1.15 | 1.08 |
| Racial Identity Importance [a] |  |  |  |
| Essential to Identity | 2.11 * | 1.94 | 1.20 |
| Important to Identity | 1.13 | 2.59 ** | 1.15 |
| Yes, Racialized Actions | 2.33 ** | 1.21 | 1.76 * |
| A Lot of Discrimination Exists | 3.44 *** | 1.15 | 1.07 |
| Yes, Experienced Discrimination | 1.23 | 1.26 | 0.98 |
| Social Pressure |  |  |  |
| Yes, Pressure from Family | 0.79 | 0.55 | 0.52 |
| Yes, Pressure from Friends | 0.67 | 0.32 * | 0.37 * |
| Yes, Pressure from Society | 2.48 * | 6.76 *** | 2.91 ** |
| None to Some White Friends | 16.41 *** | 3.70 *** | 1.96 ** |
| None to Some White Neighbors | 1.84 ** | 2.01 * | 0.94 |
| Age | 1.02 ** | 1.02 * | 0.98 ** |
| Constant | 0.00 | 0.01 | 0.75 |
| Wald chi-square (df) | 291.07 (48) |  |  |
| Pseudo $R^2$ | 0.31 |  |  |

[a] Reference category: White Only, Upper-Middle/Upper Class, Republican Party, and Not Important to Identity.
* $p < 0.05$. ** $p < 0.01$. *** $p < 0.001$.

## 3. Results

Descriptive statistics for the weighted sample are shown in Table 1. Approximately 45% of respondents selected a White only race label. The majority of respondents were female (59.08%) and claimed a middle-class social status (51.77%). Approximately 46% of respondents reported a Democrat political party affiliation. The average age of respondents was 54 years.

*3.1. Regression Results*

Table 2 presents the multinomial logistic regression results of the implications of gender, social class, and political party affiliation for race self-label choices. Table 2 indicates that among respondents in the survey, holding all other variables constant, females were 36% less likely to select a multiracial race label instead of a White only race label compared to men. At the same time, social class was not a statistically significant predictor of race self-labeling. Compared to those respondents who reported being affiliated with the Republican Party, respondents that claimed a Democratic Party affiliation were 14 times more likely to select a Black only race label and 6 times as likely to select a Hispanic only—no races label than a White only race label. In addition, respondents that claimed an Independent Party affiliation were almost 3 times more likely to select a Black only race label than a White only race label. Holding all other variables constant, for each addition year of age, respondents were 2% less likely to select a multiracial race label in comparison to selecting a White only race label.

3.1.1. Racial Identity, Discrimination, and Social Pressure

Table 3 presents the multinomial logistic regression results from three models that add one by one the factors of racial identity importance, discrimination, and social pressure to the model presented in Table 3 to examine the effect of these additional factors for race self-labeling and to investigate whether these additional factors might intercede the

implications of gender, social class, and political party affiliation for race self-labeling choices.

Mode l in Table 3 took into consideration the effects of racial identity importance and racialized actions. Respondents that claimed race as being essential to their identity were 3.5 times more likely to select a Black only race label and 2.4 times more likely to select a Hispanic only no races race label compared to a White only race label. In addition, respondents that claimed race as being important to their identity were 1.9 times more likely to select a Black only race label and 2.7 times more likely to select a Hispanic only no races race label compared to a White only race label. Similarly, respondents that indicated they had acted in a specific way to present their race were 2.4 times more likely to select a Black only race label and 66% more likely to select a multiracial label than a White only race label. When compared to the model in Table 2, Model 1 in Table 4 demonstrates that after adding the measures of racial identity, the estimate for gender became significant as a predictor of a Black only race label.

Model 2 in Table 3 took into consideration the effects of opinions on the existence of discrimination and experiences of discrimination. Respondents that believed a lot of discrimination exists towards Black and Hispanic people were 3.9 times more likely to select a Black only race label than a White only race label. Opinions on the existence of discrimination were not predictive of a Hispanic only—no races or Multiracial only race label. Compared to those who have not experienced discrimination, respondents that have experienced discrimination were 2.89 times more likely to select a Black only race label and 2 times more likely to select a Hispanic only—no races race label than a White race label. When compared to the model in Table 2, Model 2 in Table 3 demonstrates that after adding the measures of discrimination, the estimate for gender became significant as a predictor of a Black only race label.

Model 3 in Table 3 explored the effects of social pressure, specifically family pressure, friend pressure, general societal pressure, and social networks. Pressure from family members and pressure from friends to choose one race from their background were not significant predictors of race labels. In contrast, pressure from society in general to select a single race from their background resulted in respondents being 3.3 times more likely to select a Black only race label, 7 times more likely to select a Hispanic only—no races race label, and 3 times more likely to select a Multiracial race label compared to a White only race label. Moreover, respondents that indicated they had none to some White friends in comparison to having most to all White friends were 20.8 times as likely to select a Black only race label, 4.1 times as likely to select a Hispanic only—no races race label, and 96% more likely to select a Multiracial race label than a White only race label. Similarly, respondents that indicated they had none to some White neighbors were 56% more likely to select a Black only race label and 97% more likely to select a Multiracial race label than a White only race label. When compared to the model in Table 2, Model 3 in Table 3 demonstrates that after adding the measures of social pressure, social class, specifically middle class, becomes a significant predictor of race label. Specifically, respondents that identify as middle class were 53% less likely to select a Black only race labels and 65% less likely to select a Hispanic only—no races race label than a White only race label.

### 3.1.2. Gender, Social Class, Political Party Affiliation, and Additional Factors

Table 4 presents the full model that includes all study variables. Considering all factors and comparing the results to the model in Table 2 with only main independent variables, the estimate for social class, specifically middle class, became significant as a predictor of race label. Models in Table 3 that account for additional factors by introducing blocks of relevant measures one by one suggest measures of social pressure (Model 3 in Table 3) might be responsible for social class becoming a significant predictor of race label. In addition, the estimate for age became significant as a predictor for each race label option. Interestingly, for each additional year of age, respondents were 2% more likely to select a

Black only or Hispanic only—no races race label and 2% less likely to select a Multiracial race label than a White label.

## 4. Discussion

I examined whether nonracial social categories, such as gender, social class, and political party affiliation, influence race self-labels of multiracial people. In addition, I investigated whether racial identity, discrimination, and social pressure have implications for race self-labeling and might mediate the associations of nonracial social categories (i.e., gender, social class, and political party affiliation) and race self-labeling. Research on the influence of nonracial categories and race self-labeling decisions of multiracial people is growing. This study adds to the body of research and will possibly provide a starting point for further research into the social categories that influence race self-label decisions of multiracial people.

### 4.1. Gender, Social Class, and Political Party Affiliation

The results of this study indicate that the nonracial categories of gender, social class, and political party affiliation predict differences between people that claim a single minority or multiracial race self-label. When examining the three nonracial categories individually, gender was a significant predictor only for a Multiracial race label. This finding was inconsistent with expectations of prior research that indicated women would be less likely to select a single minority race label and more likely to select a multiracial race label because of negative interactions and a lack of racial validation by their single-race minority peers (Davenport 2016b; Hunter 2005; Rockquemore 2002; Rockquemore and Brunsma 2008).

Social class was associated with selecting a White only race self-label in opposition to selecting a minority only race self-label. This is right in line with prior research that indicates the category of "poor people" is mentally represented as relatively Black or minority and the category "rich people" is mentally represented as White (Lei and Bodenhausen 2017; Penner and Saperstein 2013). The change in significance for social class when all the variables were added to the model illustrates how race is negotiated to gain privilege (Bonilla-Silva 2018; Crenshaw et al. 1995; Delgado and Stefancic 2012).

Political party affiliation was also predictive of race self-labeling for a minority only race self-label. This finding provides support for prior research that highlights the racial divide of political party preference, with the majority of White people claiming a Republican Party affiliation and the majority of minority people claiming a Democrat Party affiliation (Davenport 2016a; Pew Research Center 2015; Pietraszewski et al. 2015). In addition, it highlights the mechanisms of social identity theory and critical race theory by which people self-categorize themselves into groups that elevate self-esteem, instill a sense of pride, and create situations that increase access to privilege (Crenshaw et al. 1995; Delgado and Stefancic 2012; Stets and Burke 2000; Tajfel 1982; Tajfel and Turner 1979; Turner and Reynolds 2011).

### 4.2. Racial Identity, Discrimination, and Social Pressure

In the present study, respondents that felt race was essential to their identity were more likely to select a minority only label. This is consistent with social identity theory's assertion that people self-categorize according to the values and meanings that are important to them, as this creates a sense of pride and provides a script for appropriate behavior (Tajfel 1982; Tajfel and Turner 1979).

In addition, respondents that indicated they had performed acts to intentionally present themselves as a specific race were more likely to select a Black only or Multiracial race label than a White race label. This is consistent with the literature on colorism and is most likely connected to phenotypical representations and experiences of race. The addition of a racial identity factor did not make a difference in the impact of the nonracial categories of gender, social class, and political party affiliation on race self-labeling decisions in this analysis.

The addition of a discrimination factor did not have an impact on the implications of the nonracial categories of gender, social class, or political party affiliation for racial self-labeling in this study. This was relatively consistent with prior research suggesting that encounters with racism and discrimination raised peoples' awareness of group membership with one race or another, creating two types of identity experiences, one being a person of color and the other being a multiracial person (Miville et al. 2005).

Experiences with discrimination did not significantly predict race self-labels for the overall model. This was an unexpected result that is in contrast with prior research in which multiracial people possess a fluid experience of racial identity that helps them manage the impact of discrimination. This fluidity violates the established social norms of a stable race identification and can lead to negative social interactions (Albuja et al. 2017; Crenshaw et al. 1995; Delgado and Stefancic 2012; Harris and Sim 2002). Multiracial people in turn learn or choose to enact a contextual racial presentation. One possible explanation for the present results is that adding additional factors of racial identity and social pressure reduced the impact of experiences of discrimination or was part of the process of racial identity negotiation.

Pressure from society in general illustrated the impact of negotiating race in a systemic racist society (Crenshaw et al. 1995; Delgado and Stefancic 2012). In addition, according to social identity theory, people adopt the identity of the group that they have categorized themselves as members of and in turn work to elevate the status of the group to increase pride and self-esteem (Stets and Burke 2000; Tajfel 1982; Tajfel and Turner 1979). The demands of group membership associated with family groups, friend groups, and being a member of society in general can differ in expectations and meaning, resulting in different levels of relationships and effects on racial identity.

These findings provide support for prior research indicating racial identity choices for multiracial people are contextual and often change according to the expectations of social roles and group membership (Albuja et al. 2017; Harris and Sim 2002; Rockquemore and Brunsma 2008; Saperstein and Penner 2012).

Overall, the social pressure factors had no impact on the nonracial social categories of gender or political party affiliation. However, the addition of the social pressure factors did impact social class. This is consistent with prior research indicating that economic and residential affluence associated with White ethnicity "whitens" racial identification (Davenport 2016b). Multiracial people with the ability to "pass" as a White person may be more inclined to self-identify as White to obtain access to privileges and resources (Bonilla-Silva 2018; Bratter 2018; Crenshaw et al. 1995; Davenport 2016b; Delgado and Stefancic 2012; Hunter 2005; Rockquemore and Brunsma 2008).

*4.3. Age*

In the study, I controlled for age. The findings are consistent with prior research that suggests as people age, they are more likely to adapt their identity to meet the societal expectations of race that are perpetuated and maintained within a White supremacist racial system. Years of being corrected for violating the social norms and expectations of race impact the selection of a racial identity (Albuja et al. 2017; Bonilla-Silva 2018; Crenshaw et al. 1995; Delgado and Stefancic 2012; Harris and Sim 2002).

Overall, the results of the present study indicate that the nonracial categories of gender and political party affiliation remain independent predictors of race self-labeling even after taking into consideration the additional factors of racial identity, discrimination, and social pressure. The findings also suggest a more consistent predictive ability for Black only and Hispanic only—no races race labels. Moreover, these findings illustrate the complexity of race, the social constructions of race, and the impact of White supremacy on race self-labeling decisions.

*4.4. Limitations of the Study*

This study had a few limitations. First, the dataset used for this study excluded genders other than male and female. The survey did not provide possible responses for other genders, such as non-binary and transgender people. This limited people to responses that may not accurately represent their gender identity and potentially changes the meanings associated with the nonracial category of gender. Another limitation is the data capture a single point in time, and responses can change over time in reaction to social and life circumstances. Moreover, the use of survey data limits respondents to preselected responses and may not capture all possible responses. Finally, survey data capture choices but not the motivations behind the choices.

*4.5. Future Research*

There are three areas in which future research would benefit the examination of the influence of nonracial categories on race self-labeling choices of multiracial people. First, because the present study shows that age might be an important predictor of racial self-labeling, a longitudinal study, preferably similar to the Survey of Multiracial Americans, which can record and analyze race self-labeling choices over a lifetime, might be able to highlight whether and how race self-labeling decisions might be contextual and change over time as people mature and group memberships change (Albuja et al. 2017; Morning and Saperstein 2018; Pew Research Center 2015; Rockquemore and Brunsma 2008; Shih and Sanchez 2009). A longitudinal examination could capture and correlate social and life course changes and the impact they may have on race self-labeling choices.

Second, the present study demonstrates that additional factors such as discrimination, social pressure, and racial identity importance do not completely explain, if at all, why people from certain groups defined by gender, social class, and/or political party affiliation select specific racial self-labels. Future research should explore other factors that might make a difference in how people identify themselves in terms of race. In particular, a qualitative approach would be beneficial. In-depth interviews with an intersectional approach could help explore the distinctive meanings behind race self-labeling choices. This would also allow for a more detailed application of intersectionality theory (Hill Collins and Bilge 2016). The interviews could further examine the ways in which the nonracial categories intersect and inform race self-labeling choices. This form of inquiry could confirm, refute, or enhance the relationships identified from quantitative studies such as the current study.

Relatedly, it might be beneficial in the future to look at other social categories or social structures that could potentially influence race self-labeling decisions of multiracial people. For example, the U.S. educational system portrays the ideology that all citizens are entitled to an education. Yet, the education that people receive is often unequal, especially for people of color (Bonilla-Silva 2018; Dhillon-Jamerson 2018; Harvey et al. 2017). People of color are disproportionately targets of prejudice in the educational system. They are more likely to be graded more severely, receive harsher punishments, and receive less academic instruction (Bonilla-Silva 2018; Dhillon-Jamerson 2018; Harvey et al. 2017). This occurs in an educational system that is promoted as equal fair and a basic right to citizens. The choices of multiracial respondents on the survey could be influenced by their educational experience depending on the treatment they received as based on their racial classification during their education. Adding additional nonracial categories and social structures to an examination of race self-labeling choices could provide additional support needed to identify areas where social change is needed.

**5. Conclusions**

In this study, I reviewed and expanded on the growing body of research on the multiracial population that focuses on identifying and describing nonracial categories important to shaping racial identities. Specifically, I utilized a national survey of U.S. adults administered by the Pew Research Center to investigate how social identities defined by

nonracial categories, such as gender, social class, and political party affiliation, influence the race self-labels chosen by multiracial people in the United States.

The study found that gender, social class, and political party affiliation are potentially important predictors of race self-labeling choices of multiracial people. These nonracial categories remain significant predictors after adding the factors of racial identity, discrimination, and social pressure. The results for social class and political party affiliation reinforce the actuality that a pervasive racial hierarchy and social stratification system is embedded within U.S. society (Lei and Bodenhausen 2017; Strmic-Pawl 2014). Relatedly, political party affiliation is tied to a sense of culture, values, and moral convictions, and the results imply a considerable political and racial divide (Davenport 2016a; Hochschild and Weaver 2007; Weaver 2012).

Assessing the race self-labeling decisions of multiracial people provides insight on how nonracial categories inform the contextual nature and personal understandings of race in the United States (Davenport 2016b; Shih and Sanchez 2009). Furthermore, it illustrates the influence of socialization within in a racist social system designed to perpetuate and maintain White supremacy. Thus, this study adds to the expanding and crucial body of research on race self-labeling decisions.

**Funding:** This research received no external funding.

**Institutional Review Board Statement:** Ethical review and approval were waived for this study due to the use of a secondary data set.

**Informed Consent Statement:** Not applicable.

**Data Availability Statement:** The data used in this study may be found on the Pew Research Center website: http://www.pewsocialtrends.org/dataset/survey-of-multiracial-adults/ (accessed on 16 October 2021).

**Conflicts of Interest:** The author declares no conflict of interest.

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
