# Peer review of "Multiracial Race Self-Labeling Decisions: The Influence of Gender, Social Class, and Political Party Affiliation"

_socsci, doi:10.3390/socsci11020087_

Round 1
Reviewer 1 Report
This paper uses data from the Pew Research Center’s Survey of Multiracial Americans to explore if and how gender, social class, and political party affiliation, as well experiences with and knowledge of discrimination and age influence self-identification as multiracial. This research helps us to better understand when people choose to identify as multiracial. The author clearly explains previous scholarship in this area and how their work builds on this knowledge. This research also identifies important lines of future work in this area.
I really only have one question and possible suggestion for the author. The variables used to identify both existence of and experience with discrimination are collapsed into dichotomous variables in this analysis. But, if I am understanding them correctly there was considerable variation in the response categories before they were collapsed. I’m wondering if the author tried creating these variables as an index of experience with discriminiation and knowledge of discrimination? And, does this change the results at all? It seems like it is possible that more experience with discrimniation and/or more knowledge of it could change racial identification in ways that are currently being obscured by the dichotomous variables. This could address some of the unexpected findings around discrimination, as well. If I misunderstood, or if there is a reason these variables are coded this way, I don’t think this comment alone weakens the paper at all.
I strongly recommend this paper for publication. I think the author has some excellent contributions and does a wonderful job making their case.
Author Response
I started with the same sentiment and wanted to use an index but was guided by an advisor to use the dichotomous option instead. To be honest, I cannot remember their reasoning and I failed to return to that option after the initial project was complete. This was originally my masters thesis project.
Reviewer 2 Report
Ms. ID: socsci-1444561
Reviewer Comments
This paper reports on a secondary data analysis of Pew Research Center data collected from multiracial Americans in 2015. Because multiracial people still receive relatively limited attention in research, the focus of this investigation is on an important, albeit understudied, group. The large dataset is a strong point of this work, as studies of large multiracial samples are still rare. The investigation of predictors of racial self-labeling is grounded in CRT and SID. The paper does a particularly nice job interpreting the findings in the context of these theories. Despite these impressive features, the paper has several areas that would benefit from further improvement.The major areas of concern include:
- Lack of conceptual distinction between racial self-labeling, racial identity, and racial identity salience
- Underdeveloped section on racial socialization. Mismatch between how the literature defines racial socialization and how it is operationalized in this study.
- Potential effect (as a control variable) of multiracial definition (first-generation multiracial vs. multi-generation multiracial) is not considered
- Need to integrate recent work on racial socialization of multiracial people
Specific suggestions for revision include:
- Explain critical race theory more holistically. The brief description focused only on the tenet of interest convergence. Is there a particular reason for this? What about all the other tenets of critical race theory?
- Harris (2016) developed the critical multiracial theory (MultiCrit), which was informed by critical race theory. This multiracial focused theory might be a more appropriate framework to ground the current study.
- Furthermore, given the focus on other social identity dimension such as social class and gender, an intersectional framework should be incorporated into the theoretical grounding of this work.
- Proofread the entire paper to correct incorrect spellings (e.g., ‘Rockquemore is misspelled as “Rockquemoore” throughout the paper), and parenthetic citations do not seem to consistently follow a specific style guideline (it’s unclear which style guide the paper follows)
- The discussion of multiracial people’s ability and choice to enact their whiteness to gain power and access to privileged resources seems one sided. It neglects an entire and growing segment the multiracial population, namely minority-minority multiracial individuals (e.g., Latinx-Black biracial or Asian-Black biracial). If this paper is focused on multiracial individuals whose background includes white, then this should be made explicit at the beginning of the paper.
- In the discussion of the Miville et al. study, consider integrating much more recent research on multiracial microaggressions, monoracism vs. racism (see Atkin & Yoo, 2019; Atkin et al., 2021; Nadal et al., 2013)
- Clearly define what colorism is (i.e., preferential treatment given to lighter-skinned people over darker-skinned individuals) on page 6.
- The paragraph on racial socialization should be developed into a more nuanced discussion. In its current state it does not really address racial socialization strategies used in contemporary multiracial families. There are two emerging lines of inquiry that can inform and should be integrated into this paragraph based on Atkin et al.’s work on racial-ethnic socialization and familial support of multiracial experiences. Atkin identified ‘multiracial identity socialization’ as a salient racial socialization strategy used in families of multiracial youth. This strategy specifically focused on racial identification. See Atkin et al’s work published in 2021. The section could be expanded and focused around this particular socialization strategy and replace the paragraph on neighborhood/geographic location, which does not directly relate to racial socialization.
- The conceptual distinction between racial self-labeling and racial identity is unclear. This needs to be clarified and made explicit early on in the paper. Rockquremore, Brunsma, and Delgado (2009) be useful to inform this distinction (see their discussion of racial identity, racial identification, and racial categorization). Likewise, given that the question used to inform the racial identity variable really only gets at racial identity salience, relevant literature should be discussed to contextualize the focus on identity salience.
- The two items that were used to operationalize racial socialization do not tap racial socialization. They measure racial-composition of friend group and neighborhood.
- Using White and male as reference groups for race and gender as this approach reifies white supremacy and patriarchy (see scholarly writing on how analytic decisions can perpetuate the racial status quo).
- Based on the way racial identity was measured, it appears that the variable captures racial identity salience, specifically. There is literature on this dimension of racial identity. It is only one dimension of racial-ethnic identity (see Umana-Taylor et al.’s article published in Child Development in 2014 or Bob Selleck’s work focused on Black racial identity dimensions (salience, centrality, public vs. private regard etc.). There are also scholars who focus on racial identity salience among multiracial people (e.g., the work of Gaither, Pauker, Wong, or Douglass). Some of this work should be cited in the relevant section of the literature review.
- Given the expansive definition of multiraciality used in this study, it would be helpful to know the breakdown of the sample by these subgroups (e.g., more than two races vs. multiracial based on their parents’ race(s) vs. multiracial based upon grandparents’ race(s). It would be important to examine whether racial identity, socialization, and discrimination vary between these subgroups. The experiences of first-generation multiracial people differ qualitatively from those of multi-generation multiracials (see Song, 2019 or Atkin & Yoo, 2019). If differences are found, this variable should be controlled for in the analyses.
- To clarify, please rephrase the sentence “Social class was associated with in opposition of selecting a minority only race self-label.” (p. 17, line 635)
Author Response
I have uploaded a document with my responses.
Thank you for everything, very helpful.
